# Mini-Review on the Synthesis of Furfural and Levulinic Acid from Lignocellulosic Biomass

Zhiwei Jiang , Di Hu, Zhiyue Zhao, Zixiao Yi, Zuo Chen and Kai Yan *

School of Environmental Science and Engineering, Sun Yat-sen University, Guangzhou 510275, China; jiangzhw8@mail.sysu.edu.cn (Z.J.); hudi5@mail.sysu.edu.cn (D.H.); zhaozhy36@mail2.sysu.edu.cn (Z.Z.); yizx3@mail2.sysu.edu.cn (Z.Y.); chenz6@mail2.sysu.edu.cn (Z.C.)
* Correspondence: yank9@mail.sysu.edu.cn; Tel.: +86-20-39277362

**Abstract:** Efficient conversion of renewable biomass into value-added chemicals and biofuels is regarded as an alternative route to reduce our high dependence on fossil resources and the associated environmental issues. In this context, biomass-based furfural and levulinic acid (LA) platform chemicals are frequently utilized to synthesize various valuable chemicals and biofuels. In this review, the reaction mechanism and catalytic system developed for the generation of furfural and levulinic acid are summarized and compared. Special efforts are focused on the different catalytic systems for the synthesis of furfural and levulinic acid. The corresponding challenges and outlooks are also observed.

**Keywords:** biomass; catalytic conversion; furfural; levulinic acid; mechanism; synthesis

## 1. Introduction

Biomass, the largest carbon-neutral renewable source on Earth, is considered a promising alternative to our dependence on limited fossil fuel sources and the associated environmental issues [1–6]. Combustion and anaerobic digestion are adopted extensively to treat biomass; however, these techniques are inefficient and possess problems. Therefore, the valorization of biomass into conventional fuels and chemicals has influenced a large number of research efforts [7,8]. The conversion of biomass to low-molecular weight compounds requires an effective decomposition method. In addition, the various biomass-derived chemicals can be converted into highly valuable products [9–11], which offers competitive superiority against fossil-based products. A wide range of technologies (e.g., gasification, pyrolysis, catalysis and aqueous phase processing) are successfully employed to convert biomass into fine chemicals and biofuels [12,13].

As DOE/NREL reports, the top 10 valuable chemical candidates generated from biomass were identified, and an updated top 10 + 4 list was proposed by Bozell et al. [14]. Among the various biomass-based chemicals, furfural and levulinic acid (LA) are considered two significant platform chemicals in lignocellulosic biorefineries to produce a wide variety of liquid fuels [15–18]. They can be directly obtained through the dehydration reaction of six- and five-carbon sugars (e.g., fructose and xylose). The production pathways of furfural and levulinic acid from alternative carbon sources are illustrated in Figure 1.

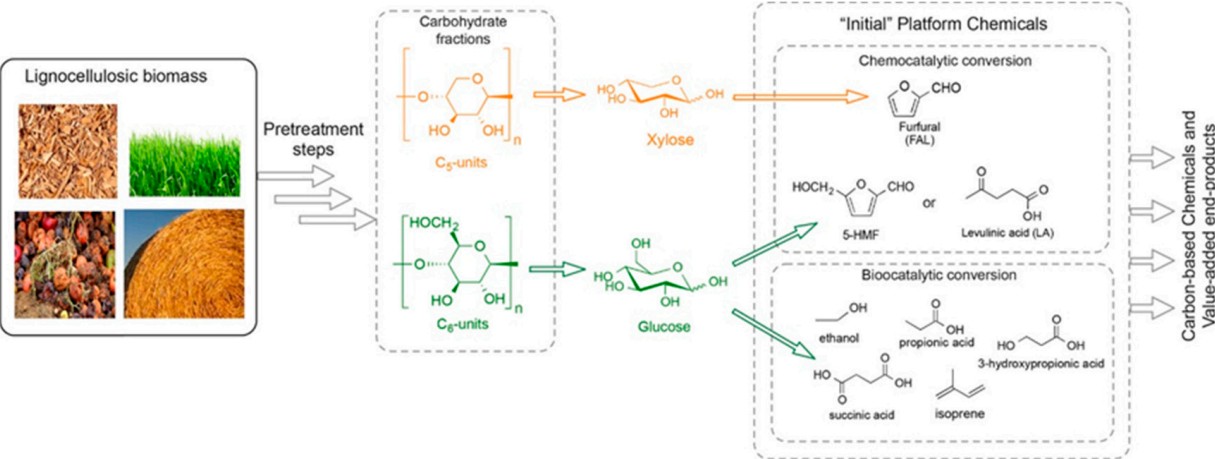

**Figure 1.** Biorefining scheme for the upgrading of lignocellulosic biomass into value-added platform chemicals and liquid fuels [19]. Reproduced with permission from Ref. [19].

A vast number of chemicals (e.g., furfuryl alcohol, furanones et. al) can be yielded from furfural that has a global production of ~300 kton/y. For example, furfuryl alcohol derived from furfural accounts for about 62% of the global furfural market and 75% of the U.S. market. Moreover, more than 80 value-added chemicals, as fossil fuel alternatives, can be synthesized using furfural as a feedstock via different reactions for the aldehyde group and aromatic ring in furfural [20]. The valorization approach of furfural is a highly attractive topic.

Levulinic acid is regarded as one of the top platform chemicals for the biorefinery. To be economically competitive, LA is generally acquired from multiple steps of the hydrolysis of biomass via 5-hydroxymethylfurfural intermediate [17]. Conversion of LA can augment the categories of chemicals from renewable biomass, which is part of an important expanded field of biomass conversion. As a sustainable platform chemical, LA with a bifunctional gamma-keto-carboxylic acid is exploited in a wide range of applications such as polymer precursors, pyrrolidinones, pharmaceutical intermediates and fuel additives. LA also reacts with phenol or aldehydes to produce various important chemicals [21]. The above discussion demonstrates that furfural and LA hold impressive market potential. However, the outcome of this fascinating goal is dependent on the large-scale and sustainable production of furfural and LA with high efficiency and at acceptable prices. In this review, we concentrate on the mechanism and various well-studied production routes of furfural and LA from different biomass resources. Some novel homogeneous (deep eutectic solvents, ions liquids) and heterogeneous catalysts (carbon-based acids, clay, oxides, zeolites, cation exchange resin, heteropoly acids, metal-organic frameworks) in various systems for the production of furfural and LA are also discussed, especially from native biomass (such as wheat straw, corn stover, sugarcane bagasse and so on). Challenges for large-scale applications and areas that need improvement are also highlighted.

## 2. Mechanism for the Formation of Furfural

The preparation of furfural is often studied through the degradation of pure xylose in an acidic environment [22–27]. Recently, there is a growing trend to utilize a complex carbohydrate and a native biomass as the starting reactant. However, the exact mechanism is not mature. Two popular mechanisms are proposed. The first one is the production of furfural from C5-sugar (e.g., xylose and arabinose) through the 1,2-enediol intermediate (Figure 2). The rate-limited step of this mechanism is the formation of 1,2-enediol intermediate. Some halide ions (e.g., $Cl^-$, $Br^-$) can tremendously improve the rate of enolization and the following dehydration reaction in an acidic environment [28].

**Figure 2.** Catalytic formation of furfural from pure xylose [29]. Reproduced with permission from Ref. [29].

The other mechanism was based on acyclic dehydration [30]. It is often considered that the hydrogen protons ($H^+$) can protonate with the hydroxyl group attached to a carbon atom (first step in Figure 3). A molecular water was lost, and the positive charged carbon atom was produced (second step in Figure 3). A double bond was prone to produce through the two electrons from a neighboring C-O bond (third step in Figure 3), the breakage of C-O bond and hydrogen atom transfer within the molecule was in the fourth step. In the fifth step, another water was released when the $H^+$ protonated the hydroxyl oxygen. A stable ring structure was generated in the sixth step and the final furfural was formed through the 1,4-elimination reaction in the seventh step. Other mechanisms of the furfural formation are also reported. Antal et al. proposed that the furanose intermediate 2,5-anhydroxylose, which can be produced by an attack of $H^+$ on O-2 in xylose, was the key step to form furfural through dehydration [31]. This protonation of the O-2 position in xylose was also proved through NMR spectroscopy and density functional theory (DFT) calculations [32].

**Figure 3.** Acyclic dehydration mechanism from xylose to furfural [29]. Reproduced with permission from Ref. [29].

### 3. Production of Furfural from Biomass

*3.1. Homogeneous Catalysts for the Furfural Production*

Quaker Oats in 1921 successfully utilized the aqueous sulfuric acid ($H_2SO_4$) to produce a 40–50% yield of furfural [33]. After this, modified technologies were developed and tested to improve the final furfural yield. Homogeneous Lewis acids and Brønsted acids are applied for the furfural formation from biomass-based feedstocks. However, mineral acids (e.g., $H_2SO_4$, HCl) are often utilized in single phase systems during the commercial processes, causing some difficulties in the recovery of furfural from the reaction system and the operation or handing of the corrosive mineral acid. Binder et al. [34] found transition metal ions (e.g., Cr (II) or Cr (III)) as cocatalysts and HCl as the main catalyst for the transformation of xylose and xylan in an N,N-dimethylacetamide (DMAc)/LiCl solvent system. $CrCl_3$ can efficiently catalyze xylan to form furfural with a 63% yield of at 200 °C in ionic liquids under microwave-assisted heating. This catalytic system is also good for the conversion of real biomass corn stalk, rice straw, and pinewood into furfural [35]. Ionic liquids including both Lewis acidic and Brønsted acidic sites can be utilized as catalysts to convert biomass into furfural [15]. Small amounts of $[C_4SO_3Hpy][BF_4]$ ionic liquids in the $THF/H_2O$ solvent system efficiently dehydrated xylose to furfural under microwave heating at 180 °C for 1 h with as high as an 85% yield. In this catalytic system, agricultural lignocellulosic waste-derived syrup as a feedstock, produced a45% furfural yield with a longer reaction time (4 h) [36]. Recently, deep eutectic solvents (DESs) received increasing attention due to several advantages (lower cost and toxicity, ease of synthesis etc.) over ionic liquids [37]. Many researchers reported that DESs can act as reaction mediums and provide Brønsted acidity to convert biomass into furfural [38]. Choline chloride-oxalic acid-based DESs were used as a solvent and a catalyst to produce furfural with a 26.3% yield from oil palm fronds at 100 °C for 135 min [39]. To enhance the furfural yield, the addition of a second phase solvent as an extractant of furfural is a feasible strategy. Wang et al. developed a green butanone-water solvent system to efficiently convert biomass into furfural. A higher furfural yield (56%) was obtained from xylose at 140 °C. Furthermore, xylan and corn stalk were also treated under similar conditions to achieve furfural with yields of 46% and 36%, respectively [40]. Water-soluble heteropoly acids (HPAs) are also utilized as homogeneous catalysts to generate furfural. Dias et al. reported that $H_3PW_{12}O_4$ (PW), $H_4SiW_{12}O_{40}$ (SiW) and $H_3PMo_{12}O_{40}$ (PMo) as HPAs are homogeneous catalysts at 140 °C for the transformation of xylose into furfural under different solvent systems, and they showed tremendous catalytic differences. The PW HPAs examined, displayed the highest furfural yield around 63% in DMSO [41]. More homogeneous catalytic systems were developed and utilized for the generation of furfural, as summarized in Table 1. Homogeneous catalysts often presented high activity and selectivity, however, the future attention to the efficient separation of furfural from the catalytic system may require more effort.

**Table 1.** Production of furfural from carbohydrate substrates over different homogeneous catalysts.

| No. | Substrate | Catalyst | Reaction Conditions | Solvent | Conversion (%) | $Y_{furfural}$ (%) | Ref. |
|---|---|---|---|---|---|---|---|
| 1 | Xylose | $H_2SO_4$ (3.6 mg/mL) | 240 °C, 60 min | $H_2O$ | — | 78 | [42] |
| 2 | Xylose | HCl (4 mg/mL) | 200 °C, 10 min | $H_2O$ | 95 | 64 | [43] |
| 3 | Xylose | $AlCl_3$ (1.5 w/v%) | 180 °C, 30 min | ChCl:EG/acetone | 83.3 | 75 | [44] |
| 4 | Xylose | $AlCl_3$ (2 mg/mL) | 140 °C, 25 min | ChCl:citric acid/MIBK | 99.8 | 73 | [45] |
| 5 | Xylose | HCl (0.1 mol/mol) | 170 °C, 50 min | $H_2O$/MIBK | — | 85 | [46] |
| 6 | Xylose | $H_2SO_4$ (10 w/w%), NaCl (0.24 g/mL) | B.T., 5 h | $H_2O$/toluene | 98% | 83 | [47] |
| 7 | Xylose | Formic acid (10 g/L) | 180 °C, 6.5 h | $H_2O$ | 94.9 | 74 | [48] |
| 8 | Xylose | HCl (0.25 mol/L), NaCl (36%) | 170 °C, 20 min | $H_2O$/SBP | 95 | 78 | [49] |
| 9 | Xylose | HCl (0.1 M), ($CrCl_3$ 6 mM) | 145 °C, 2 h | $H_2O$/toluene | 95.8 | 76 | [50] |

**Table 1.** *Cont.*

| No. | Substrate | Catalyst | Reaction Conditions | Solvent | Conversion (%) | $Y_{furfural}$ (%) | Ref. |
|---|---|---|---|---|---|---|---|
| 10 | Alginic acid | $CuCl_2$ (0.29 mmol) | 220 °C, 1 min, MW | $H_2O$/MIBK | — | 31 | [51] |
| 11 | Alginic acid | $H_3PW_{12}O_{40}$ (10 mg) | 180 °C, 30 min | $H_2O$/THF | — | 33.8 | [52] |
| 12 | Xylan | $AlCl_3$ (0.25 mmol) | 170 °C, 10 s | [BMIM]Cl | — | 84.8 | [53] |
| 13 | Xylan | $AlCl_3$ (2 mg/mL) | 140 °C, 35 min | ChCl:citric acid/MIBK | 99.8 | 69 | [45] |
| 14 | Flax shives | HCl (pH = 1.12) | 180 °C, 20 min | $H_2O$ | — | 72.1 | [54] |
| 15 | Wheat straw | HCl (pH = 1.12) | 180 °C, 20 min | $H_2O$ | — | 48.4 | [54] |

Note: B.T: Boiling temperature; ChCl: Choline chloride; EG: Ethylene glycol; MIBK: Methyl isobutyl ketone; BMIM: 1-butyl-3-methylimidazolium; SBP: 2-Sec-butylphenol; MW: Microwave-assisted heating.

### 3.2. Heterogeneous Catalysts for Furfural Production

Heterogeneous catalysts may reduce the corrosion of the equipment and are relatively easy to recycle, reducing the cost. Some heterogeneous acid catalysts developed for furfural production are carbon acids, clay, oxides, zeolites, cation exchange resin, heteropoly acids and metal organic frameworks (MOF) [22,55–57].

Carbon-based solid acid catalysts are extensively used in substantial applications because of their high thermal stability and low cost. Zhang et al. [58] reported that Carbon-$SO_3H$ was employed as a catalyst for converting xylose and corn stalk into furfural in $\gamma$-valerolactone (GVL) with high yields of 78.5% and 60.6%, respectively. In order to separate the used carbon solid acid, magnetic carbon-based solid acid was also synthesized and applied for furfural production from the biomass [59]. Sulfonated clay and zeolite were also used for the production of furfural. A green sulfonated palygorskite solid acid catalyst (PAL-$SO_3H$) in a GVL–water mixture exhibited a high catalysis activity and good recyclability with a furfural yield of 87% from xylose at 180 °C for 1 h [60]. H-ZSM-5 zeolite as a catalyst in a GVL solvent efficiently converted corn cob into furfural with a 71.7% yield at 190 °C for 60 min. $HSO_3$-ZSM-5 showed a higher furfural yield (89%) in the THF/$H_2O$ cosolvent [61]. In addition, Gupta et al. [62] developed amorphous $Nb_2O_5$ with water-compatible Bronsted and Lewis acid sites as a reusable catalyst for the formation of furfural from xylose. As high as a 65.9% yield of furfural was achieved in biphasic water and a toluene solvent at 100 °C for 3 h. Heteropoly acid exhibited promising a catalytic performance for furfural production with low cost and strong acidity. Recently, mesoporous bi-metallic heteropoly acid salt (SnCsPW) was employed as a heterogeneous solid catalyst for the conversion of xylose into furfural in the DMSO/$H_2O$ solvent. SnCsPW with a Sn/Cs molar ration of 0.625: 0.5 exhibited the highest catalytic activity with a furfural yield of 63% at 200 °C for 3 h [63]. An MOF with a large specific surface area, high porosity, and diverse structures was also utilized for the biomass valorization. MIL-101(Cr)-$SO_3H$ with both Lewis acid and Brønsted acid sites was developed to convert xylose into furfural in a biphasic cyclopentyl methyl ether (CPME)/$H_2O$-NaCl solvent system (Figure 4)). As high as a 70.8% furfural yield and 97.8% xylose conversion were obtained at 170 °C for 3 h in this catalytic system. Table 2 summarized more heterogeneous catalytic systems for the production of furfural.

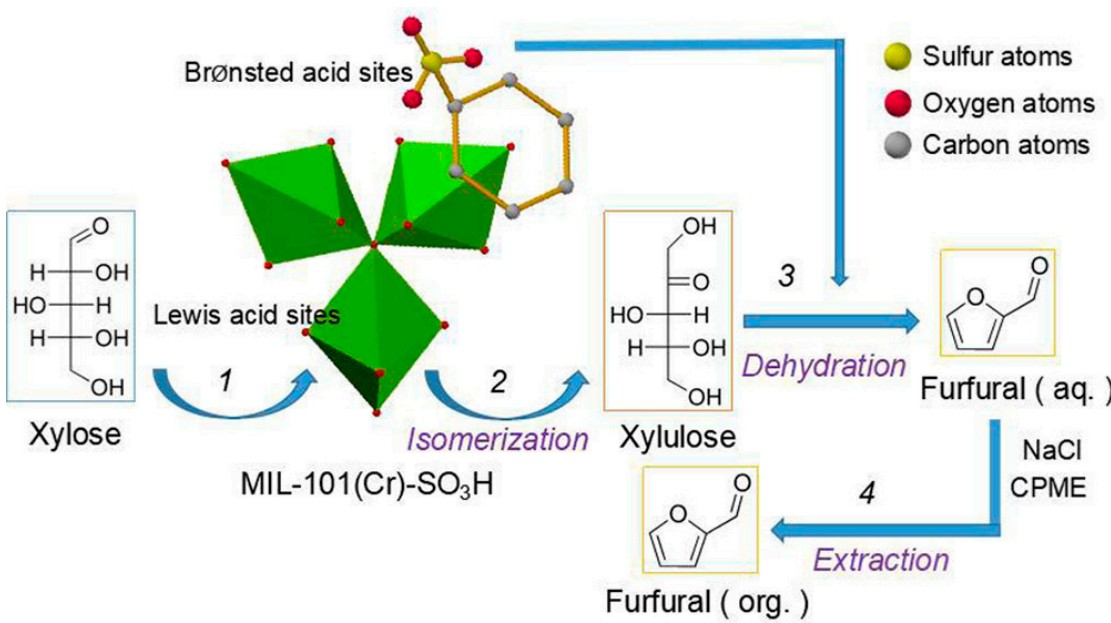

**Figure 4.** Production of furfural from xylose in the presence of the MIL-101(Cr)-SO₃H catalyst [64]. Reproduced with permission from Ref. [64].

**Table 2.** Production of furfural from carbohydrate substrates over different heterogeneous catalysts.

| No. | Substrate | Catalyst | Reaction Conditions | Solvent | Conversion (%) | $Y_{furfural}$ (%) | Ref. |
|-----|-----------|----------|---------------------|---------|---------------|--------------|------|
| 1 | Xylose | Nb₂O₅ (100 mg) | 120 °C, 3 h | Toluene/H₂O | >99 | 72 | [62] |
| 2 | Xylose | Nafion117 (1.82 wt%) | 150 °C, 2 h | DMSO | 91 | 60 | [65] |
| 3 | Xylose | Sn₀.₆₅Cs₀.₅PW (0.08 g) | 200 °C, 3 h | DMSO/H₂O | 98 | 63 | [63] |
| 4 | Xylose | g-CN-SO₃H (50 mg) | 100 °C, 30 min | H₂O | —— | 95 | [66] |
| 5 | Xylose | g-CN-SO₃H (50 mg) | 100 °C, 25 min | DMSO | —— | 95 | [66] |
| 6 | Xylose | Carbon-SO₃H (50 mg) | 170 °C, 30 min | GVL/H₂O | 99.5 | 75.1 | [67] |
| 7 | Xylose | Carbon-SO₃H (0.2 g) | 170 °C, 30 min | GVL | —— | 78.5 | [58] |
| 8 | Xylose | SnP@MIL-101(Cr) (0.1 g) | 150 °C, 3 h | Toluene/H₂O | 93.2 | 86.7 | [68] |
| 9 | Xylose | MIL-101(Cr)-SO₃H (0.27 g) | 170 °C, 3 h | CPME/NaCl-H₂O | 97.8 | 70.8 | [64] |
| 10 | Xylose | CaCl₂/γ-Al₂O₃ (0.65 g) | 150 °C, 50 min | Toluene/H₂O | 100 | 55 | [69] |
| 11 | Xylose | MIL-OTS-0.5 (200 mg) | 170 °C, 3 h | Toluene/H₂O | 94.6 | 62.6 | [70] |
| 12 | Xylose | PAL-SO₃H (50 mg) | 180 °C, 60 min | GVL/H₂O | 90.6 | 87 | [60] |
| 13 | Xylose | Form-SO₃H (0.1 M acid sites) | 170 °C, 10 min | Toluene/H₂O | —— | 70–80% | [71] |
| 14 | Xylose | CST (200mg) | 170 °C, 30 min | Toluene/H₂O | 82.5 | 62.4% | [72] |
| 13 | Xylan | SO₄²⁻/Sn-MMT (5 g/g) | 160 °C, 90 min | 2-MTHF/NaCl-H₂O | —— | 77.4 | [73] |

Note: GVL: γ-valerolactone; 2-MTHF: 2-methyltetrahydrofuran; DMSO: dimethyl sulfoxide; CPME: cyclopentyl methyl ether; THF: tetrahydrofuran; PAL-SO₃H: sulfonated palygorskite; CST: carbonized and sulfonated teff straw.

In general, several issues exist in the production of furfural: (i) Limited furfural yields in the range of 45% to 55% due to side reactions. Native biomass or complex saccharide mixtures can be utilized as substrates instead of pure xylose. (ii) The use of mineral acid led to the corrosion toward the equipment. (iii) Difficulty in catalyst recycles and complex regeneration associated with high cost. Simplified recovery of solvent and product also require more efforts. (iv) Improving the feedstock conversion and product selectivity requires more effort.

## 4. Mechanism of the Formation of Levulinic Acid

With two functional groups (carbonyl and carboxyl group), numerous aspects can be upgraded into many value-added chemicals and biofuels. The formation of LA from native

biomass received much attention. However, the complex pathway, the mechanism for the formation of LA, requires more effort. The production of LA is extensively investigated, revealing that the actual mechanism faces a great challenge. Recently, some advanced tools were utilized to elucidate a reaction intermediate and kinetics [74–77]. It is generally believed that hydroxymethyl furfuran (HMF) is the key intermediate which is rehydrated to generate LA and formic acid. Antal et al. [78] reported that fructose was converted to HMF through a series of cyclic furan intermediates (Figure 5, b pathway). Moreover, Moreau et al. [79,80] and Lamport et al. [81] proposed that fructose also converted into HMF through an enediol pathway (Figure 5, a pathway) in the presence of an alkaline catalyst. The enediol as an intermediate was proposed during the isomerization process of glucose to fructose. The further rehydration of HMF to LA was associated with the addition of one molecular water to the $C_2$ –$C_3$ bond of the furan ring, leading to the formation of LA and formic acid (see Figure 5). Recently, a great deal of mechanistic studies focused on acid-catalyzed transformation of hexose sugars to HMF and subsequently to LA using a nuclear magnetic resonance spectroscopy to reveal intermediate products and thus determine the reaction pathways of the LA formation [76,78,82–85].

**Figure 5.** Possible dehydration mechanisms for formation of HMF (**up**) acyclic route; (**down**) cyclic route and the formation of LA from HMF.

## 5. Production of Levulinic Acid from Biomass

### 5.1. Homogeneous Catalysts for the Levulinic Acid Production

Due to the high selectivity, strong Brønsted acid (e.g., HCl, $H_2SO_4$) catalysts are often utilized in the conversion of biomass to produce LA. Mulder et al. [86] in the 1840s successfully utilized the mineral acids (HCl, $H_2SO_4$) catalyst in a high temperature to produce LA from saccharide. After this, some strategies such as the introduction of the second solvent and microwave irradiation was employed to improve the LA yield over Brønsted acid catalysts [17]. Dumesic and coworkers developed a biphasic acid aqueous and organic solvent reactor system for efficient synthesis of LA from pure saccharide and

complex saccharide [87]. Organic acids as a catalyst were also utilized for the generation of LA from lignocellulosic biomass [88]. Lewis acid (such as Fe, Cr, Sn ions) accelerates the isomerization of glucose to fructose [89–91], and thus increases the LA yield. A mixed-acid system with both Lewis acids and Brønsted acids ($CrCl_3-H_3PO_4$) was developed to produce LA from glucose, which was proven to possess a strong synergic catalytic activity with 49.8% yield of LA [92]. In recent years, studies have increased for ionic liquids as a solvent or catalyst for LA production from lignocellulose [18,93,94]. Ren et al. [95] reported that a sulfonated ionic liquid ($[C_3SO_3Hmim]HSO_4$) exhibited high catalysis activity and good recyclability with an LA yield of 55% from cellulose by microwave-assisted synthesis with at 160 °C for 30 min. Moreover, Li et al. [96] found that phosphotungstic acid could improve the transformation of glucose and cellulose to LA in a deep eutectic solvent. A 98.9% yield of LA from glucose was achieved at 195 °C for 80 min, while 88.6% was obtained from cellulose for 120 min at the same reaction temperature.

Recently, sawdust, corn stover, sugarcane bagasse, municipal waste and pulp sludge received more attention due to their low-cost as they are treated as waste and their potentially high value as the source of biofuels and value-added chemicals. Lappalainen et al. [97] studied the valorization of pulp industry waste biomass into LA with $H_2SO_4$ and Lewis acids as catalysts in a microwave reactor. As high as a 56% LA yield could be obtained at 180 °C for 60 min in 0.3 M $H_2SO_4$ and 7.5 mM $CrCl_3$. In addition, potato peel waste and Cortinarius armillatus were also efficiently transformed into LA under the similar reaction conditions with the yields of 49% and 62%, respectively [98]. Bamboo biomass, rubber wood, and palm oil frond were directly converted in LA using dicationic acidic ionic liquids, with a higher yield of LA (47.5%) from bamboo biomass at 110 °C for 60 min [99]. It was efficient for lignocellulose by pretreatment to break the lignin fraction, and thus enhance the accessibility of lignocellulose. Moreover, glucosamine, a monomer of chitosan, also can be a feedstock for LA generation. Kim et al. [100] studied the conversion of glucosamine to LA with 33.8 mol% at 200 °C and 1 M $H_2SO_4$ for 15 min. Table 3 summarizes different homogeneous catalysts for the synthesis of LA from lignocellulose.

**Table 3.** Production of LA from lignocellulose and biomass over different homogeneous catalysts.

| No. | Substrate | Catalysts | Reaction Conditions | Solvent | Conversion (%) | $Y_{LA}$ (%) | Ref. |
|---|---|---|---|---|---|---|---|
| 1 | Glucose | [IL-SO$_3$H][Cl] (400 mg), NiSO$_4$ (1.8 wt%) | 155 °C, 5 h | Water | 99.92 | 56.4 | [101] |
| 2 | Glucose | Phosphotungstic acid (30 mg/mL) | 195 °C, 80 min | GVL/water | 100 | 98.9 | [96] |
| 3 | Cellulose | H$_2$SO$_4$ (1 wt%) | 180 °C, 1 h | Sulfolane/water | 100 | 72.5 | [102] |
| 4 | Cellulose | Phosphotungstic acid (30 mg/mL) | 195 °C, 120 min | GVL/water | 100 | 88.6 | [96] |
| 5 | MCC | SnCl$_2$: HCl (1:1, 0.17 M) | 198 °C, 5 h | Water | 100 | 63.5 | [89] |
| 6 | DSB | [EMim][HSO$_4$] (4 g) | 100 °C, 7 h | MIBK | 100 | 62 | [103] |
| 7 | DSB | [BMMim][BF$_4$] (4 g) | 100 °C, 7 h | MIBK | 100 | 54.2 | [104] |
| 8 | Microalgae | H$_2$SO$_4$ (0.8 M) | 180 °C, 40 min | Water | 100 | 39.9 | [105] |
| 9 | Sorghum | [BMIM][HSO$_4$] (16.7%) | 180 °C, 30 min | Water | 100 | 11.7 | [106] |
| 10 | Hemp hurd | HCl (0.1 g/g) | 120 °C, 12 h | [EMIM]Cl | 100 | 59 | [107] |
| 11 | Chlorella vulgaris | HCl (0.95 M) | 170 °C, 30 min | Water | 100 | 39.3 | [108] |
| 12 | Lignocellulose | p-TsOH (0.95 M) | 162 °C, 64 min | Water | 100 | 57.9 | [88] |
| 13 | Corncob | SnCl$_4$ (0.768 M) | 180 °C, 1 h | Water | —— | 64.6 | [109] |
| 14 | Corn stalk | FeCl$_3$ (0.5 M) | 180 °C, 40 min | Water | —— | 48.9 | [110] |
| 15 | Corncob | p-TsOH (1.1 M) | 180 °C, 70 min | Water | —— | 61.3 | [111] |
| 16 | Pine | CrCl$_3$ (0.1 M) | 180 °C, 4 h | GVL/water | —— | 78 | [112] |

Note: GVL: γ-valerolactone; DSB: depithed sugarcane bagasse; MCC: microcrystalline cellulose; [EMim][HSO$_4$]: 1-ethyl-3-methylimidazolium hydrogen sulfate; [BMMim][BF$_4$]: 1-butyl-2,3-dimethylimidazolium tetrafluoroborate; MIBK: methyl isobutyl ketone; DESs: deep eutectic solvents; [EMIM]Cl: 1-ethyl-3-methylimidazolium chloride; p-TsOH: toluene sulfonic acid.

### 5.2. Heterogeneous Catalysts for the Levulinic Acid Production

The studies for LA synthesis over environmentally friendly heterogeneous acid catalysts gained increased attention in recent years [17,113]. Some types of heterogeneous catalysts (such as carbon-based materials, oxides, zeolites, cation exchange resin, heteropoly acids and MOF) were developed.

Carbon-based solid acid was studied in the generation of LA. Boonyakarn et al. [114] reported a new hybrid catalytic system using both Bronsted acid (HTCG-SO$_3$H) and Lewis acid (CrCl$_3$) for the LA synthesis from cellulose with a 40% yield at 200 °C for 5 min. The regulation of Bronsted/Lewis acid ratio can improve the LA yield from biomass. A high LA yield of 64.2% was achieved from glucose over mesoporous Fe-NbP with the appropriate Bronsted/Lewis acid ratio (1.32) at 180 °C in 3 h [115]. Zeolites are crystalline oxide materials with well-defined pores and adjustable Bronsted acidity. HZSM-5, SAPO-18, SBA-15, and HY are widely applied in acid hydrolysis. Wei et al. [116] utilized the Cr/HZSM-5 catalyst and the hydrolysis of cellulose with a high LA yield of 64% at 180 °C for 3h in aqueous media. The catalyst can be reused at least four times without significant reduction in the reaction activity. Ion-exchange resins such as Nafion, Dowex, amberlyst and solid Bronsted acid are used for LA production. Amberlyst 36, a strongly acidic cation exchange resin, was applied for LA production from vegetable waste in DMSO/water solvent. As high as a 17% LA yield was obtained at 120 °C in 5 min with the aid of microwave heating [117]. The type of salt in aqueous solution was also affected the LA yield from sugar. A higher yield of LA (74.6%) was obtained from glucose at 110 °C for 24 h in NaCl aqueous solution compared to other salt (KCl, CaCl$_2$, Na$_2$CO$_3$, and Na$_2$SO$_4$) aqueous solutions [118]. Recently, heteropolyacids (HPAs) as heterogeneous acid catalysts were reported to achieve cellulose conversion with high efficiency. Lewis metal-substituent phosphotungstic acids MH$_n$PW$_{11}$O$_{39}$ (MHPW, M=Cu, Sn, Cr, Zn, Fe, Ti, Zr, M=Ti, Zr, oxygen is 40) were used for the LA formation. TiH$_5$PW$_{11}$TiO$_{40}$ showed the most activity with the highest cellulose conversion and the highest LA yield (70.9%) in methyl isobutyl ketone (MIBK)/water under microwave-assisted heating [119]. MOFs and MOF-based materials are extensively designed in biomass transformation. Different metal oxides (CeO$_2$, Ga$_2$O$_3$, and CoO) were loaded in Zr-MOF material (UiO-66) to adjust its catalysis activity and acid strength, which were employed in the transformation of cellulose to LA. Ga$_2$O$_3$-UiO-66 exhibited a maximum LA yield of 32% at 240 °C for 6h in aqueous solution [120]. In addition, insoluble lysine functionalized phosphotungstic acid was encapsulated on MIL-100 (Fe) as a bifunctional solid catalyst (Lys-PM$_2$) for the production LA from glucose, as shown in Figure 6 [121]. A 57.9% yield of LA was achieved at 150 °C for 9 h and the catalysis activity basically remained unchanged after four times of reuse. Table 4 summarizes different heterogeneous catalysts for the generation of LA from lignocellulose.

Mineral acid (e.g., HCl, H$_2$SO$_4$) catalyzed sugars to form LA with higher yields. However, mineral acid was often associated with the corrosion toward the equipment, and difficult handling. The feedstocks (such as carbohydrates) for the LA preparation often competes with the food chain. Future research can develop a novel catalytic system in which the reviewable and non-edible feedstocks can be employed and selectively transformed into LA. We believed that the heterogeneous acidic and porous catalysts can be more promising.

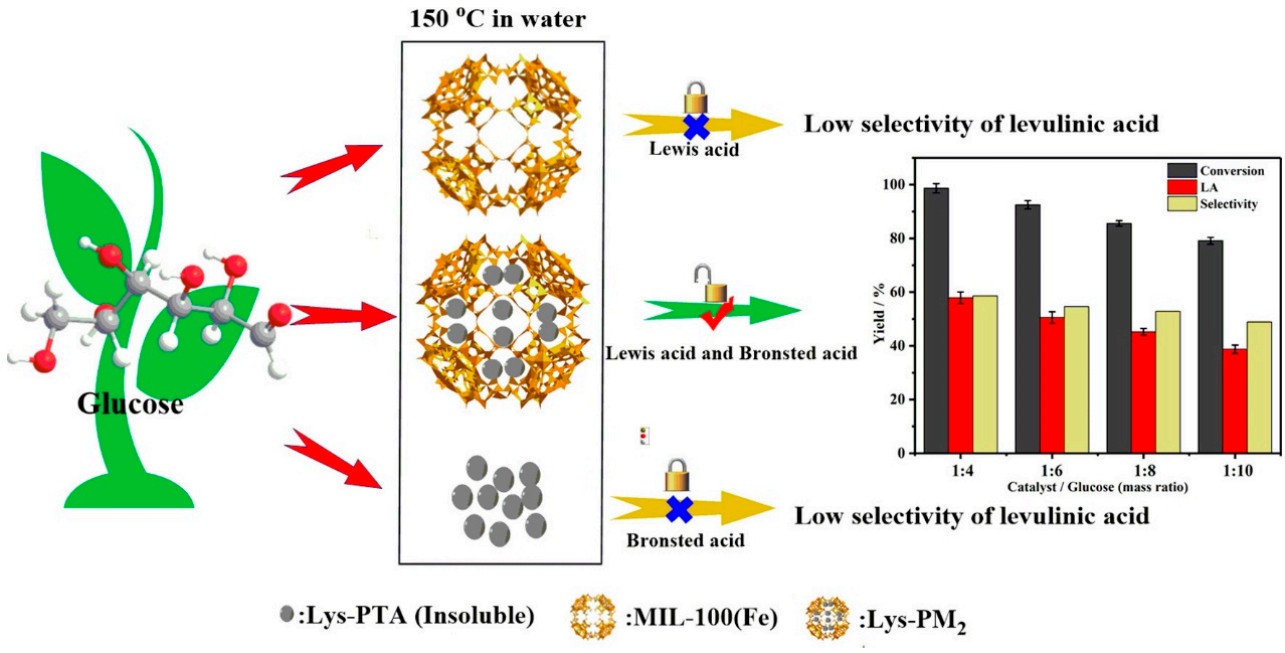

**Figure 6.** Production of furfural from glucose in the presence of the Lys-PM$_2$ catalyst [121]. Reproduced with permission from Ref. [121].

**Table 4.** Production of LA from lignocellulose and biomass over different heterogeneous catalysts.

| No. | Substrate | Catalysts | Reaction Conditions | Solvent | Conversion (%) | Y$_{LA}$ (%) | Ref. |
|---|---|---|---|---|---|---|---|
| 1 | Fructose | H-Resin (30 wt%), NaCl (10 wt%) | 110 °C, 24 h | Water | >98 | 74.6 | [118] |
| 2 | Fructose | S-beta zeolite (3 wt%) | 160 °C, 7 h | Water | 98.15 | 43.5 | [122] |
| 3 | Fructose | HSO$_3$-SBA-15 (200 mg) | 180 °C, 5 h | Water | 100 | 30 | [123] |
| 4 | Fructose | Dowex 50×8-100 (50 mg) | 120 °C, 24 h | Water/GVL | 99 | 72 | [124] |
| 5 | Glucose | Cr/HZSM-5 (0.75 g) | 180 °C, 3 h | Water | 100 | 64.4 | [116] |
| 6 | Glucose | Fe-NbP (50 mg) | 180 °C, 3 h | Water | 98.9 | 64.2 | [115] |
| 7 | Glucose | H-Resin (60 mg), NaCl (10 wt%) | 145 °C, 24 h | Water | 100 | 74.6 | [118] |
| 8 | Glucose | Al$_{4/3}$SiW$_{12}$O$_{40}$ (0.45 g) | 180 °C, 2 h | Water | 100 | 50.4 | [125] |
| 9 | Glucose | Lys-PM$_2$ (25 mg) | 150 °C, 9 h | Water | 98.7 | 57.9 | [121] |
| 10 | Cellulose | Ga$_2$O$_3$-UiO-66 (2.5%) | 240 °C, 6 h | Water | —— | 32 | [120] |
| 11 | Cellulose | ChH$_4$PWTi (3.09 mmol) | 130 °C, 8 h | MIBK/water | 93.8 | 76.1 | [119] |
| 12 | Cellulose | HTCG-SO$_3$H 5 wt%, CrCl$_3$ (0.015 M) | 200 °C, 5 min | Water | —— | 40 | [114] |
| 13 | Sorghum stems | Mn$_3$O$_4$ (2%), phosphoric acid (40%), H$_2$O$_2$ (30%) | 130 °C, 10 h | Water | —— | 26.6 | [126] |
| 14 | Corn stover | SAPO-18 (0.3 g) | 190 °C, 80 min | Water | —— | 70.2 | [127] |
| 15 | Rice Husk | Mn$_3$O$_4$/ZSM-5, H$_3$PO$_4$ (10%), H$_2$O$_2$ (2%) | 130 °C, 8 h | Water | —— | 39.8 | [128] |
| 16 | Bagasse | Sn-MMT /SO$_4^{2-}$ (0.1 g), saturated NaCl | 180 °C, 3 h | Water | —— | 62.1 | [129] |

Note: S-beta zeolite: sulfonated beta zeolite; MIBK: methyl isobutyl ketone; GVL: γ-valerolactone; Lys-PM$_2$: lysine functionalized phosphotungstic acid.

## 6. Conclusions

We conducted a comprehensive review of the mechanisms and current technologies of furfural and levulinic acid production from various biomass-derived feedstocks. The development of environmentally friendly technologies for the generation of valuable chemicals and alternative fuels from renewable biomass is gaining tremendous attention. A rapid acceleration of studies aimed at developing highly efficient catalysts and the reaction systems for the efficient transformation of biorenewable feedstocks to furfural

and LA. Furfural and LA are very promising biorefinery platform chemicals and they are often synthesized from a wide variety of different biomass feedstocks with a minimal environmental load. Monosaccharides such as pure glucose or fructose as the starting substrate offer the highest product yields. The use of complex polysaccharides, and even native lignocellulosic materials, often exhibits much lower yields, and are receiving more efforts to develop various advanced catalytic systems.

Mineral acid catalysts are utilized extensively in the lab-scale, but in larger-scale processes there are huge challenges to producing products with high purity and without degradation, recovering and recycling the catalyst, and the safety hurdles. Heterogeneous catalysts are easier to recover and recycle, but tend to achieve relatively lower product yields than homogenous catalysts. However, the development of non-toxic, cheap, high-performance, and easily recyclable catalysts is promising. The acid properties and pore geometry in heterogeneous catalysts are reported to affect the yield of furfural and LA. The process parameters also affect the yield of furfural and LA. By studying catalytic systems, which catalyst properties control bond scission, what role secondary metals play in the reactive chemistry for catalysts, how molecular functionality influences reaction pathways and what kinetic requirements can maximize activity and selectivity. The use of operando tools can provide more insight into the reaction process, especially for catalyst deactivation. For example, the operando synchrotron-based X-ray techniques can detect the changes in the surface and active sites of the catalyst, revealing the active centers for the formation of furfural and LA. The operando Raman spectra can gauge the reaction intermediates to verify the mechanisms of the formation of furfural and LA. The cost-effective separation of furfural and levulinic acid from the reaction system faces a lot of challenges. One-step conversion of original biomass to furfural and levulinic acid without complicated pre-treatment and the isolation of intermediates is highly desirable. Future efforts are needed in the design of robust catalyst and solvent systems. The development of a biphasic system combined with a solid catalyst containing the acid function is also very interesting for future studies, especially in industrial application.

**Author Contributions:** Writing—original draft preparation, Z.J., K.Y.; writing—review and editing, D.H., Z.Z., Z.Y., Z.C.; visualization, D.H., Z.Z., K.Y.; supervision, K.Y.; project administration, K.Y.; funding acquisition, Z.J., K.Y. All authors have read and agreed to the published version of the manuscript.

**Funding:** This research was funded by National Key R&D Program of China (2018YFD0800703, 2020YFC1807600), National Ten Thousand Talent Plan, National Natural Science Foundation of China (22078374, 21776324, 21905309), Key-Area Research and Development Program of Guangdong Province (2019B110209003), Guangdong Basic and Applied Basic Research Foundation (2019B1515120058, 2020A1515011149), the Fundamental Research Funds for the Central Universities (19lgzd25) and Hundred Talent Plan (201602) from Sun Yat-sen University.

**Institutional Review Board Statement:** Not applicable.

**Informed Consent Statement:** Not applicable.

**Acknowledgments:** Not applicable.

**Conflicts of Interest:** The authors declare no conflict of interest.

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
