# Peer review of "Mini-Review on the Synthesis of Furfural and Levulinic Acid from Lignocellulosic Biomass"

_processes, doi:10.3390/pr9071234_

Round 1

Reviewer 1 Report

Manuscript ID: processes-1250504

The mini-review submitted by Zhiwei Jiang et. al deals with biomass valorization to platform chemicals, concretely levulinic acid and furfural. Although the topic is highly relevant, the language quality, presentation format and the lack of overall statements/conclusions drive me to reject the draft. The draft needs deep review:

  • English must be deeply corrected, the text is full of mistakes. In some cases I don’t even understand what the authors want to say (specially in pages 1 to 6).
  • Furfural part: the presentation format is far to be interesting for the reader. The authors must structure this mini-review properly, not only giving yield values over the tested catalysts. Indeed, the huge amount of results in the text results hard to follow. The authors must introduce properly the topic, the overall scenario and then describe appropriately each section. Writing a mini-review is not just summarize all the furfural yields, the authors must highlight similar behaviours, trends, results… the lack of overall statements/conclusions is decisive. The latter also for levulinic acid part.
  • Levulinic acid part: the presentation and included contents are better than those of furfural but still this part must be deeply improved.

Author Response

Thanks for your kindly suggestions. The review has been deeply corrected in the revised manuscript. Some results, overall scenario, and conclusions have been corrected or added in the revised manuscript. Some references were added or corrected. Some contents in Table were also corrected.

Reviewer 2 Report

The paper entitled “ Mini-review on the synthesis of furfural and levulinic aid from lignocellusic biomass” with the reference processes-1250504 for Processes mdpi journal deals with the the valorisation of renewable biomass into furfural (FUR) and levulinic acid (LA), as important platform chemicals as intermediates to a wide range of valuable bioproducts with potential to substitute fossil fuels. However, a poor discussion on FUR applications to give answers to these fossil fuels alternatives is presented. Instead, the authors gave an extreme importance to the two FUR mechanisms production, reprinting the mechanisms already discussed in detail in their reference 26, not presenting any new advantageous comment on them. The same applies for the section about LA. I also had a quick search on the ISI Web of knowledge and realized that there are some new reviews on furfural production from lignocellulosic biomass (e.g. , , ), which makes me wonder what does this review add? This should be clarified for the readers to know in what sense is this review different from all the others published and what new insights does it give to complement the new reviews published.

Besides, The English language is poor, making it difficult to follow the authors´ ideas. Furthermore, the title or subtitles have missing letters or letters changed, which shows some lack of care. Some concerns are listed below. For these reasons, I find that this manuscript does not meet the standards of the journal Processes.

Line 24 the authors say that gasification and pyrolysis have been successful employed to convert biomass but in Line 19 they wrote that combustion was inefficient.

Line 26- What is DOF/NREL report? The authors do not clarify.

Line 35- The authors wrote “ furans et al”?

Line 37-38- confusing sentence that should be clarified: “Moreover, many value-added chemicals were classified by the application and business model also can be synthesized using furfural as the feedstocks via different techniques”.

Line 58-60-  The authors wrote: “The formation of 1,2-enediol is the rate-limited step and it was reported that the presence of halide ions 58 (e.g., Cl−, Br−) in the acidic environment would greatly increase the rate of enolization and the followed dehydration reaction [25]”.  This does not seem to be mentioned in referred ref 25.

Line 120-121- The authors wrote: “Heteropoly acid exhibited promising catalytic performance for furfural production with low cost, strong acidity, and strong oxidizing capability”.

What is the interest in the oxidizing capability for the production of furfural?

The authors reproduce one scheme for the production of furfural from xylose in the presence of a MOF with the permission of the mentioned authors. Although I do not see what’s the interest in reproducing one very specific schemewhen the authors are comparing results from the literature of a wide range of heterogeneous catalysts and conditions.

Table 2 (starting in Line 142) is very incomplete. The authors do not specify the criteria used for the choice of the catalysts they chose to be compared. They compared mixed oxides (Nb2O5), resins (Nafion 117), carbon catalysts (CN-SO3H), MOFs (SnP@MIL-101(Cr) and MIL-101(Cr)-SO3H), but they do not mention zeolites or mesoporous silicas which are studied such as the recently paper: . The authors did not include other types of heterogeneous catalysts such as sulfonated foams (DOI: 10.1016/j.cattod.2020.12.009), or other catalysts derived from biomass such as lignin carbohydrate based catalysts (). The authors mentioned the catalyst Nb2O5 from ref 56 but did not cite other more recent papers that used Nb2O5 catalysts in the dehydration of xylose such as  (DOI:) and . An update of the state of the art is recommended.

Line 160-161- The authors wrote: “HMF is possibly not the intermediate for LA formation. LA formation is not catalyzed through Brönsted acids or bases [78]”.  If HMF may not be intermediate of LA why show the mechanism in Figure 5? In this case they should have given a justification for this sentence and present an alternative mechanism.

Line 168- Confusing discussion. After writing Line 160-161, the authors then write “Due to the high selectivity, strong Brønsted acids (e.g. HCl, H2SO4) catalysts are often used in biomass hydrolysis to produce LA”, which is inconsistent with their previous sentence in Lines 160-161 where they mentioned that LA is not catalysed by acids neither bases.

Line 218-219- Unfinished sentence: “Some Lewis metal-substituent phosphotungstic acid MHnPW11O39 (MHPW, M=Cu, Sn, Cr, Zn, Fe, Ti, Zr, M=Ti, Zr, oxygen is 40)” .

Line 224-225- “In addition, insoluble lysine functionalized phosphotungstic acid was encapsulated on MIL-100 (Fe) as a bifunctional solid catalyst (Lys-PM2) for the conversion of glucose to LA, as shown in Figure 6 [114]”. I think it is not necessary to copy the exact figure of the original paper because this is only one example of one work for the conversion of glucose to LA.

Line 232-233- confusing sentence that should be clarified: “However, mineral acid often associated with the corrosion toward the equipment, the difficult handling, and the  carbohydrate often competes with the food chain”.

On Table 2, the meanings of S-Beta zeolite and Lys-PM2 should be specified as footnote.

Author Response

The paper entitled “ Mini-review on the synthesis of furfural and levulinic aid from lignocellusic biomass” with the reference processes-1250504 for Processes mdpi journal deals with the the valorisation of renewable biomass into furfural (FUR) and levulinic acid (LA), as important platform chemicals as intermediates to a wide range of valuable bioproducts with potential to substitute fossil fuels. However, a poor discussion on FUR applications to give answers to these fossil fuels alternatives is presented. Instead, the authors gave an extreme importance to the two FUR mechanisms production, reprinting the mechanisms already discussed in detail in their reference 26, not presenting any new advantageous comment on them. The same applies for the section about LA. I also had a quick search on the ISI Web of knowledge and realized that there are some new reviews on furfural production from lignocellulosic biomass (e.g. , , ), which makes me wonder what does this review add? This should be clarified for the readers to know in what sense is this review different from all the others published and what new insights does it give to complement the new reviews published.

Reply: Thanks for your kindly suggestion. For the discussion on FUR applications, the sentence of “Moreover, many value-added chemicals were classified by the application and business model also can be synthesized using furfural as the feedstocks via different techniques” in Line 37-38 has been corrected into “Moreover, more than 80 value-added chemicals, as fossil fuels alternatives, also can be synthesized using furfural as the feedstocks via different types of reaction for aldehyde group and aromatic ring in Furfural”.

Some other mechanisms for FUR production and LA were added in the revised manuscript. The sentences of “Besides, some other mechanisms of the furfural formation are also reported. Antal et. al. proposed that furanose intermediate 2,5-anhydroxylose, which can be produced by an attack of H+ on O-2 in xylose, is the key step to form furfural through dehydration (Carbohydr. Res. 1991, 217, 71-85). This protonation of O-2 position in xylose was also proved through NMR spectroscopy and density functional theory (DFT) calculations (J. Phys. Chem. A 2006, 110, 11824-11838).” and were added in the corresponding location of the revised manuscript.

In this review, we mainly concentrated the mechanism of the furfural and LA formation as well as the yield and selectivity of furfural and LA, especially from native biomass, over different homogeneous and heterogeneous catalysts in various system. In addition, some recent advances in the formation of furfural and LA have also been discussed in this review. “Some homogeneous and heterogeneous catalysts in various system for the production of furfural and LA are also discussed, especially from native biomass.” has replaced “The current technologies for their productions are identified and their potential application as well as the fuel properties are discussed.”

Besides, The English language is poor, making it difficult to follow the authors´ ideas. Furthermore, the title or subtitles have missing letters or letters changed, which shows some lack of care. Some concerns are listed below. For these reasons, I find that this manuscript does not meet the standards of the journal Processes.

Reply: The whole manuscript has been corrected carefully. The title or subtitles have been revised.

Line 24 the authors say that gasification and pyrolysis have been successful employed to convert biomass but in Line 19 they wrote that combustion was inefficient.

Reply: Thanks for your kindly suggestion. Combustion was inefficient, which just produced small amount of thermal energy. Gasification and pyrolysis of biomass can prepare fine chemicals and biofuels. This description is reasonable.

Line 26- What is DOF/NREL report? The authors do not clarify.

Reply: Thanks for your kindly suggestion. DOF should be DOE (Department of Energy), NREL is abbreviation of National Renewable Energy Laboratory.

Line 35- The authors wrote “ furans et al”?

Reply: Thanks for your kindly suggestion. You are right. The phrase “and furans” in Line 35 should be cancelled.

Line 37-38- confusing sentence that should be clarified: “Moreover, many value-added chemicals were classified by the application and business model also can be synthesized using furfural as the feedstocks via different techniques”.

Reply: Thanks for your kindly suggestion. “Moreover, many value-added chemicals were classified by the application and business model also can be synthesized using furfural as the feedstocks via different techniques” has corrected into “Moreover, more than 80 value-added chemicals, as fossil fuels alternatives, also can be synthesized using furfural as the feedstocks via different types of reaction for aldehyde group and aromatic ring in Furfural”.

Line 58-60- The authors wrote: “The formation of 1,2-enediol is the rate-limited step and it was reported that the presence of halide ions 58 (e.g., Cl−, Br−) in the acidic environment would greatly increase the rate of enolization and the followed dehydration reaction [25]”. This does not seem to be mentioned in referred ref 25.

Reply: Thanks for your kindly suggestion. You are right. The reference should be Carbohydr. Res. 2011, 346 (11), 1291-1293. And the referred reference has been corrected.

Line 120-121- The authors wrote: “Heteropoly acid exhibited promising catalytic performance for furfural production with low cost, strong acidity, and strong oxidizing capability”. What is the interest in the oxidizing capability for the production of furfural?

Reply: Thanks for your kindly suggestion. The phrase “and strong oxidizing capability” should be cancelled.

The authors reproduce one scheme for the production of furfural from xylose in the presence of a MOF with the permission of the mentioned authors. Although I do not see what’s the interest in reproducing one very specific scheme when the authors are comparing results from the literature of a wide range of heterogeneous catalysts and conditions.

Reply: Thanks for your kindly suggestion. This Figure emphasizes the effects of Lewis acid and Brønsted acid on the conversion xylose into furfural.

Table 2 (starting in Line 142) is very incomplete. The authors do not specify the criteria used for the choice of the catalysts they chose to be compared. They compared mixed oxides (Nb2O5), resins (Nafion 117), carbon catalysts (CN-SO3H), MOFs (SnP@MIL-101(Cr) and MIL-101(Cr)-SO3H), but they do not mention zeolites or mesoporous silicas which are studied such as the recently paper: . The authors did not include other types of heterogeneous catalysts such as sulfonated foams (DOI: 10.1016/j.cattod.2020.12.009), or other catalysts derived from biomass such as lignin carbohydrate based catalysts (). The authors mentioned the catalyst Nb2O5 from ref 56 but did not cite other more recent papers that used Nb2O5 catalysts in the dehydration of xylose such as (DOI:) and . An update of the state of the art is recommended.

Reply: Thanks for your kindly suggestion. The sulfonated forms and lignin carbohydrate based heterogeneous catalysts have been added on Table 2.

Line 160-161- The authors wrote: “HMF is possibly not the intermediate for LA formation. LA formation is not catalyzed through Brönsted acids or bases [78]”. If HMF may not be intermediate of LA why show the mechanism in Figure 5? In this case they should have given a justification for this sentence and present an alternative mechanism.

Reply: Thanks for your kindly suggestion. The sentences “HMF is possibly not the intermediate for LA formation. LA formation is not catalyzed through Brönsted acids or bases [78]” should be cancelled.

Line 168- Confusing discussion. After writing Line 160-161, the authors then write “Due to the high selectivity, strong Brønsted acids (e.g. HCl, H2SO4) catalysts are often used in biomass hydrolysis to produce LA”, which is inconsistent with their previous sentence in Lines 160-161 where they mentioned that LA is not catalysed by acids neither bases.

Reply: Thanks for your kindly suggestion. Strong Brønsted acids are often used in biomass hydrolysis to produce LA. The sentences “HMF is possibly not the intermediate for LA formation. LA formation is not catalyzed through Brönsted acids or bases [78]” should be cancelled.

Line 218-219- Unfinished sentence: “Some Lewis metal-substituent phosphotungstic acid MHnPW11O39 (MHPW, M=Cu, Sn, Cr, Zn, Fe, Ti, Zr, M=Ti, Zr, oxygen is 40)” .

Reply: Thanks for your kindly suggestion. You are right. The phrase “were used for the LA formation” was added in the end of the sentence.

Line 224-225- “In addition, insoluble lysine functionalized phosphotungstic acid was encapsulated on MIL-100 (Fe) as a bifunctional solid catalyst (Lys-PM2) for the conversion of glucose to LA, as shown in Figure 6 [114]”. I think it is not necessary to copy the exact figure of the original paper because this is only one example of one work for the conversion of glucose to LA.

Reply: Thanks for your kindly suggestion. In this review, there are tow example about MOF-based catalysts for the LA formation. So, we provide one figure for that.

Line 232-233- confusing sentence that should be clarified: “However, mineral acid often associated with the corrosion toward the equipment, the difficult handling, and the carbohydrate often competes with the food chain”.

Reply: Thanks for your kindly suggestion. The sentence “However, mineral acid often associated with the corrosion toward the equipment, the difficult handling, and the carbohydrate often competes with the food chain” should be corrected into “However, mineral acid often associated with the corrosion toward the equipment and the difficult handling. The feedstocks (such as carbohydrates) for the LA preparation often competes with the food chain.”

On Table 2, the meanings of S-Beta zeolite and Lys-PM2 should be specified as footnote.

Reply: Thanks for your kindly suggestion. The meanings of S-Beta zeolite and Lys-PM2 has be specified as footnote on Table 4.

Reviewer 3 Report

The article »Mini-review on the synthesis of furfural and levulinic acid from lignocellulosic biomass« by Jiang et al. discusses the mechanism of formation for furfural and levulinic acid. It additionally discusses the production of both from biomass.

Introduction:

The introduction part surmises the topics of the manuscript well. The authors show a good understanding of the subject field. Overall, the English should be improved.

Mechanism for the formation of furfural

  1. The description for Figure 3 is a rewritten part from Ref. [29]. I fail to see the relevance of this.

Production of furfural from biomass

  1. In the tables, the authors should add a conversion value. The yield that is reported is great, but a number showing the conversion of biomass per mass of catalyst is also a relevant number.
  2. BMIM abbreviation is used in Table 1 but it is not described anywhere.
  3. In Table 2 GBL is referenced, however, it is not used in the table or the text.

Mechanism of the formation of levulinic acid

  1. This part is concise and well written.

Production of levulinic acid from biomass

  1. Table 3 should contain a conversion value and the mass of the catalyst.
  2. Table 4 should contain a conversion value and the mass of the catalyst.

Conclusion:

The conclusion could benefit from some improvement. The English should definitely be improved. The authors do a very good job pointing out some of the problems of the described processes. They do not, however, present any recommendations to future researchers. For instance, what operando techniques are available, and what do they elucidate.  

Author Response

Review (major revision):

The article »Mini-review on the synthesis of furfural and levulinic acid from lignocellulosic biomass« by Jiang et al. discusses the mechanism of formation for furfural and levulinic acid. It additionally discusses the production of both from biomass.

Introduction:

The introduction part surmises the topics of the manuscript well. The authors show a good understanding of the subject field. Overall, the English should be improved.

Reply:

Thanks for your kind suggestions. The English has been improved carefully.

Mechanism for the formation of furfural

  1. The description for Figure 3 is a rewritten part from Ref. [29]. I fail to see the relevance of this.

Reply:

Thanks for your kind suggestions. The description for Figure 3 is one mechanism for the formation of furfural form xylose through acyclic dehydration. Therefore, we believe that the description for Figure 3 related with the Pate 2 (Mechanism for the formation of furfural). This part indeed referenced Ref. [29]

Production of furfural from biomass

  1. In the tables the authors should add a conversion value. The yield that is reported is great, but a number showing the conversion of biomass per mass of catalyst is also a relevant number.

Reply:

Thanks for your kind suggestions. The conversion values and the mass of the catalyst have been added in Table 1 and Table 2.

  1. BMIM abbreviation is used in Table 1 but it is not described anywhere.

Reply:

Thanks for your kind suggestions. You are right. BMIM abbreviation was added in the Table footnote.

  1. In Table 2 GBL is referenced, however, it is not used in the table or the text.

Reply:

Thank for your kind suggestions. You are right. GBL has been cancelled.

Mechanism of the formation of levulinic acid

  1. This part is concise and well written.

Reply:

Thanks for your kind suggestions.

Production of levulinic acid from biomass

  1. Table 3 should contain a conversion value, and the mass of the catalyst.

Reply:

Thanks for your kind suggestions. The conversion values and the mass of the catalyst have been added in Table3.

  1. Table 4 should contain a conversion value, and the mass of the catalyst.

Reply:

Thank for your kind suggestions. The conversion values and the mass of the catalyst have been added in Table 4.

Conclusion:

The conclusion could benefit from some improvement. The English should definitely be improved. The authors da a very god job pointing out some of the problems of the described processes. They do not, however, present any recommendations to future researchers. For instance, what operando techniques are available, and what do they elucidate.

Reply:

Thank for your kind suggestions. The English has been improved carefully. Some examples for operando techniques were added in the revised manuscript. The sentence “For example, the operando synchrotron-based X-ray techniques could detect the changes in the surface and active sites of catalyst, revealing the active centres for the formation of furfural and LA. The operando Raman spectra could gauge the reaction intermediates to verify the mechanisms of the formation of furfural and LA.” was added in the Conclusion parts.

Round 2

Reviewer 2 Report

The submitted review entitled by “ Mini-review on the synthesis of furfural and levulinic aid from lignocellusic biomass” with the reference processes-1250504 for Processes mdpi journal has been revised by the authors which had in general satisfied the comments previously done.

However, in order to be accepted for processes, I still think that some details should be improved, such as the English language, especially in the two points pointed below:

  • The title still has an error in “aid” instead of acid. I think that the title is very important and should have no errors.
  • The sentence “ Homogeneous catalysts often presented high conversion activity and selectivity” is confusing. Conversion and activity are related with each other. Shouldn´t it be enough to say high activity and selectivity?

Author Response

The submitted review entitled by “ Mini-review on the synthesis of furfural and levulinic aid from lignocellusic biomass” with the reference processes-1250504 for Processes mdpi journal has been revised by the authors which had in general satisfied the comments previously done.

However, in order to be accepted for processes, I still think that some details should be improved, such as the English language, especially in the two points pointed below:

Reply: Thanks for your kind comments and suggestions.

The title still has an error in “aid” instead of acid. I think that the title is very important and should have no errors.

Reply: Thanks for your kind suggestions. The word “aid” in Title has been corrected into “acid”.

The sentence “ Homogeneous catalysts often presented high conversion activity and selectivity” is confusing. Conversion and activity are related with each other. Shouldn´t it be enough to say high activity and selectivity?

Reply: Thanks for your kind suggestions. The word “conversion” has been cancelled in the revised manuscript.

Reviewer 3 Report

All my comments have been addressed. Though the English still could use some work.